# A method for explaining individual predictions in neural networks

Sejong Oh

Department of Software Science, Dankook University, Youngin, Gyeonggi-do, Republic of Korea

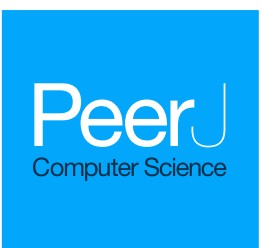

## ABSTRACT

**Background:** Recently, the explainability of the prediction results of machine learning models has attracted attention. Most high-performance prediction models are black boxes that cannot be explained. Artificial neural networks are also considered black box models. Although they can explain image classification results to some extent, they still struggle to explain the classification and regression results for tabular data. In this study, we explain the individual prediction results derived from a neural network-based prediction model.

**Methods:** The output of a neural network is fundamentally determined by multiplying the input values by the network weights. In other words, the output is a weighted sum of the input values. The weights control how much each input value contributes to the output. The degree of influence of an input value $x_i$ on the output can be evaluated as ($x_i$ · weight value $w_i$)/weighted sum. From this insight, we can calculate the contribution of each input value to the output as it flows through the neural network.

**Results:** With the proposed method, the neural network is no longer a black box. The proposed method effectively explains the predictions made by the neural network and is independent of the depth of the hidden layers and the number of nodes in each hidden layer. This provides a clear rationale for this interpretation. It can be applied to both regression and classification models. The proposed method is implemented as a Python library, making it easy to use.

## INTRODUCTION

Although machine learning has achieved considerable success in recent years, prediction models generated by high-performing machine learning algorithms, such as neural networks, remain a "black box," whereas simple algorithms, such as linear regression and k-nearest neighbors, are considered "white box" models. We do not have sufficient knowledge about how specific prediction results are obtained or the underlying principles behind the functioning of machine learning models. The functional mechanisms underlying linear regression and decision tree models are relatively easy to understand, whereas those based on neural networks and support vector machines are more complex to interpret. An increasing number of areas, including medical diagnosis, require explanations of the prediction results derived by machine algorithms. The General Data Protection Regulation is a European Union regulation on information privacy that forces

Corresponding author
Sejong Oh, sejongoh@dankook.ac.kr

agencies to provide explanations for decisions made automatically by algorithms (*Harrison & Rubinfeld, 1978*; *Regulation, 2016*). The United States legally requires financial institutions to explain the reasons for their major financial decisions, such as issuing credit cards and lending (*Korea, 2004*). This trend will only strengthen in the future.

Researchers have attempted to develop methodologies for explaining the working mechanisms of prediction models and their prediction results. Most studies focus on uncovering the role of features (variables) in a prediction model (*Das & Rad, 2020*; *Dwivedi et al., 2023*; *Manresa-Yee et al., 2021*; *Schnake et al., 2024*), and a few studies provide explanations of the prediction results. Local interpretable model-agnostic explanations (LIME) (*Ribeiro, Singh & Guestrin, 2016*; *Zafar & Naimul, 2019*; *Palatnik de Sousa, Maria Bernardes Rebuzzi Vellasco & Costa da Silva, 2019*) is a local surrogate model. It explains individual predictions by replacing the complex model with a locally interpretable surrogate model such as Lasso or a decision tree. LIME constructs a new dataset by creating perturbed samples and obtaining their corresponding predictions from the black-box model. It then trains an interpretable model on this dataset, assigning weights based on the proximity of the sampled instances to the target instance. This learned model serves as a reliable local approximation of the machine learning model's predictions, though it is not required to accurately approximate the model globally (*Molnar, 2020*). Figure 1 shows an example LIME chart. LIME offers several key benefits in the field of explainable AI (XAI). One notable advantage is its model-agnostic nature, meaning it can interpret and explain the outputs of any machine learning model, regardless of the underlying algorithm or complexity. Additionally, LIME excels at providing localized explanations. By constructing a simplified surrogate model that mimics the behavior of the original complex model for a specific instance, LIME generates case-specific explanations. This feature is particularly valuable when personalized or context-specific justifications for model predictions are required (*Safjan, 2023*).

While LIME offers numerous benefits, it also has certain drawbacks. A key challenge is its reliance on human judgment when defining the kernel function. Since the choice of kernel function and its parameters directly influence the quality of explanations, users must possess relevant domain expertise to select an appropriate configuration. Another drawback of LIME is its susceptibility to variations in input data. Because LIME generates explanations by creating slightly altered versions of an instance, even minor modifications can lead to significantly different interpretations. As a result, the consistency and stability of LIME's explanations may be compromised when inputs undergo small changes (*Safjan, 2023*).

SHapley Additive exPlanations (SHAP) (*Lundberg & Lee, 2017*; *Tan et al., 2018*; *Nohara et al., 2019*; *Rodríguez-Pérez & Jürgen, 2019*) are also used to explain individual predictions. It computes the Shapley values from coalitional game theory. In Shapley value theory, the feature values of a data instance act as players in a coalition. The Shapley values indicate how to distribute the predicted values fairly among the features (players). A player can have an individual feature value or a group of feature values. Figure 2 shows an example SHAP chart. SHAP has a solid theoretical foundation in game theory and is also a model-agnostic method. The prediction is fairly distributed among the feature values. We

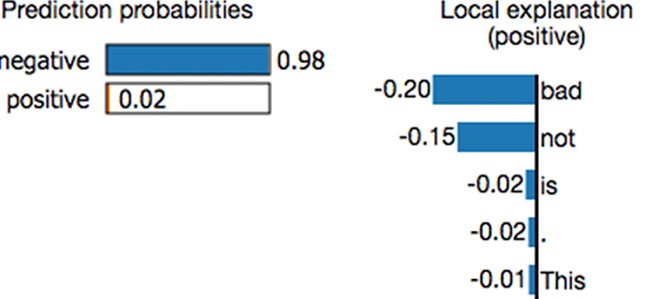

**Figure 1 Example of a LIME chart (*Ribeiro, Singh & Guestrin, 2016*).**

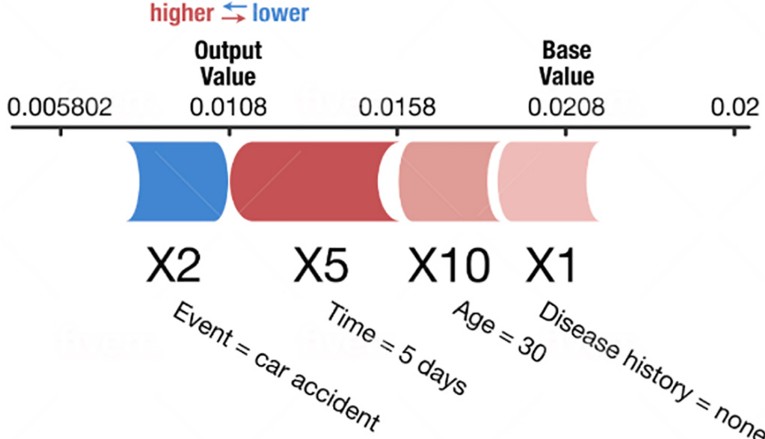

**Figure 2 Example of a SHAP chart (*Lundberg & Lee, 2017*).**

get contrastive explanations that compare the prediction with the average prediction. SHAP connects LIME and Shapley values. This is very useful to better understand both methods. It can support global model understanding such as feature importance, feature dependence, and feature interactions. SHAP also has certain drawbacks. SHAP requires long computing time. SHAP is a permutation-based method that ignores feature dependencies. By replacing feature values with values from random instances, it is usually easier to randomly sample from the marginal distribution. However, if features are dependent, this leads to putting too much weight on unlikely data points. Shapley values and SHAP can be misinterpreted, and intentionally misleading interpretations. It is possible to create intentionally misleading interpretations with SHAP, which can hide biases (*Molnar, 2020*).

In this study, we propose a new method called the NNexplainer to explain individual predictions using regression and classification models based on neural networks. Neural networks are typical black box models. Although they can explain image classification results to some extent (*Samek et al., 2021*; *Lin, Lee & Celik, 2021*; *Rguibi et al., 2022*), they still struggle to explain the classification and regression results for tabular data. In a neural network, input values flow through the network to reach the final output. The weights of a

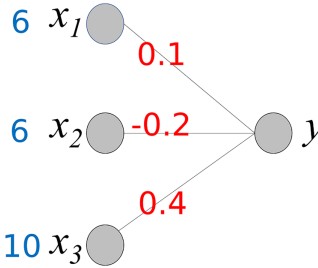

**Figure 3  Simple neural network.**               

neural network control the flow of input values similar to a faucet. If a flow is observed, the contribution of each input value to the output value (prediction) can be determined. In other words, a prediction can be resolved as the sum of portions of the input values. This served as motivation for the proposed NNexplainer. The NNexplainer mainly deals with regression models based on neural networks, but it can be easily extended to classification models. We describe the basic ideas and formal algorithms used to calculate the portion of each input value and present a composition chart to explain individual predictions. We also introduce a contribution plot to show the portions of each input value in the prediction. LIME and SHAP are compared with the proposed NNexplainer in the discussion section.

## MATERIALS AND METHODS

### Basic idea

To understand the proposed method, we consider a simple neural network, as shown in Fig. 3. It comprises three input nodes and a single output node. The activation function for the output node is $f(x) = x$. In other words, the output value is the same as the weighted sum of the weights $w$ and input $x$.

The output value $y$ is calculated using the following equation:

$$\begin{aligned}
y &= w_1x_1 + w_2x_2 + w_3x_3 \\
&= 0.1 \times 6 + (-0.2) \times 6 + 0.4 \times 10 \\
&= 0.6 + (-1.2) + 4 \\
&= 3.4.
\end{aligned} \tag{1}$$

Our goal is to determine the contribution of each input value to producing an output 3.4. As shown, the output value is the sum of $w_ix_i$ and $w_i$ is a fixed value. Therefore, we can consider $w_ix_i$ to be the contribution of $x_i$ to the output 3.4. We compute $Cx_i$, where $Cx_i$ represents the contribution of $x_i$.

$$Cx_1 = 0.1 \times 6 = 0.6 \tag{2}$$

$$Cx_2 = (-0.2) \times 6 = (-1.2) \tag{3}$$

$$Cx_3 = 0.4 \times 10 = 4 \tag{4}$$

$$y = \sum_{i=1}^{3} Cx_i = 3.4. \tag{5}$$

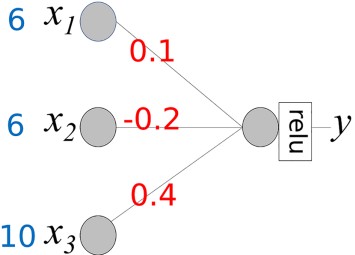

**Figure 4 Simple neural network with activation function "relu".**

This calculation process provides important insights into the interpretation of the output $y$ of the neural network. The baseline value of $y$ is 0. $Cx_1$ and $Cx_2$ increase $y$ to 4.6 (0.6 + 4), while $Cx_3$ decreases y by −1.2. As a result, the final value of $y$ is 3.3. Owing to the different corresponding weight values, the contributions of $x_1$ and $x_2$ differ even though $x_1$ and $x_2$ have the same values. Here, we can observe the role of the weight values; they control the influence of each input value on the output. Based on the weight values, we can calculate the contribution of each input value to the output $y$.

## Example 1

In the foundational idea presented above, we assume that the activation function $\phi$ for the output node is $\phi(x) = x$. However, an output node typically uses a specific activation function. In regression models, the "relu" activation function is generally used. Therefore, when calculating the contributions of the input values, the activation function should be considered. The neural network in Fig. 4 is the same as that in Fig. 3, except that the activation function is "relu." If $m >= 0$ then $relu(m) = m$, else $relu(m) = 0$.

The activation function transforms the weighted sum into an output. Equations (2)–(4) include the transformation rates. Assuming that $WS$ represents a weighted sum, output $y$ is defined as $y = relu(WS)$. The transformation rate $(TR)$ is defined as $TR = relu(WS)/WS = y/WS$. The contribution of $x_i$ is calculated using the following steps:

$$
\begin{aligned}
y &= relu(WS) \\
&= relu(3.4) \\
&= 3.4
\end{aligned}
\tag{6}
$$

$$
\begin{aligned}
TR &= y/WS \\
&= 3.4/3.4 \\
&= 1.
\end{aligned}
\tag{7}
$$

In Eq. (7), if $WS$ is 0, then, we assign 0 to $TR$.

$$
\begin{aligned}
Cx_1 &= w_1 \times x_1 \times TR \\
&= 0.1 \times 6 \times 1 = 0.6
\end{aligned}
\tag{8}
$$

$$
\begin{aligned}
Cx_2 &= w_2 \times x_2 \times TR \\
&= (-0.2) \times 6 \times 1 = (-1.2)
\end{aligned}
\tag{9}
$$

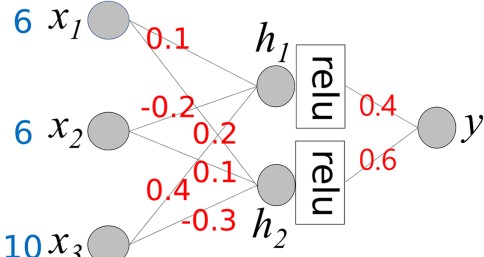

**Figure 5 A neural network that has a hidden layer.**

$$Cx_3 = w_3 \times x_3 \times TR$$
$$= 0.4 \times 10 \times 1 = 4. \tag{10}$$

Calculation of the contribution of $x_i$ can be simplified using matrix operations. Let us denote the matrices $x$, $W$, and $Cx$, as well as the function $diag(x)$:

$$x = \begin{bmatrix} x_1 \\ x_2 \\ x_3 \end{bmatrix}, \quad W = \begin{bmatrix} w_1 \\ w_2 \\ w_3 \end{bmatrix}, \quad Cx = \begin{bmatrix} Cx_1 \\ Cx_2 \\ Cx_3 \end{bmatrix}$$

$$diag(x) = \begin{bmatrix} x_1 & 0 & 0 \\ 0 & x_2 & 0 \\ 0 & 0 & x_3 \end{bmatrix}.$$

Then,

$$Cx = W^T \bullet diag(x) \times TR$$
$$= \begin{bmatrix} w_1 x_1 TR \\ w_2 x_2 TR \\ w_3 x_3 TR \end{bmatrix}, \tag{11}$$
$$= \begin{bmatrix} 0.6 \\ -1.2 \\ 4 \end{bmatrix}$$

$$y = \sum_{i=1}^{3} Cx_i = 3.4.$$

## Example 2

The simple neural network in Example 1 has no hidden layers. In a deep neural network, computing the contribution of each input value to the final output is complex. Equation (11) must be calculated repeatedly for each layer. Consider the neural network shown in Fig. 5, which includes a hidden layer. The activation function for the hidden layer is "relu," while the final output node uses the activation function $f(x) = x$.

For simple calculation, we pre-calculate TR:

$$TR_{h1} = 3.8/3.8 = 1 \qquad //TR \text{ for } h_1 \tag{12}$$

$$TR_{h2} = 0/-1.2 = 0 \qquad //\text{TR for } h_2 \tag{13}$$

$$TR_{out} = 1 \qquad //\text{TR for } y. \tag{14}$$

We also define several matrices and a new operation denoted by $\otimes$:

$$W_0 = \begin{bmatrix} 0.1 & 0.2 \\ -0.2 & 0.1 \\ 0.4 & -0.3 \end{bmatrix}, \; W_1 = \begin{bmatrix} 0.4 \\ 0.6 \end{bmatrix}, \quad TR_0 = \begin{bmatrix} TR_{h1} \\ TR_{h2} \end{bmatrix} = \begin{bmatrix} 1 \\ 0 \end{bmatrix}$$

$$\begin{bmatrix} x_{11} & x_{12} & x_{13} \\ x_{21} & x_{22} & x_{23} \end{bmatrix} \otimes \begin{bmatrix} z_1 \\ z_2 \end{bmatrix} = \begin{bmatrix} x_{11}z_1 & x_{12}z_1 & x_{13}z_1 \\ x_{21}z_2 & x_{22}z_2 & x_{23}z_2 \end{bmatrix}.$$

We first calculate $Cx$ for $h_1$ and $h_2$ in Step 0.

$$
\begin{aligned}
Cx_{Step0} &= W_0{}^T \bullet diag(x) \otimes TR_0 \\[4pt]
&= \begin{bmatrix} 0.1 & -0.2 & 0.4 \\ 0.2 & 0.1 & -0.3 \end{bmatrix} \bullet \begin{bmatrix} 6 & 0 & 0 \\ 0 & 6 & 0 \\ 0 & 0 & 10 \end{bmatrix} \otimes \begin{bmatrix} 1 \\ 0 \end{bmatrix} \\[4pt]
&= \begin{bmatrix} 0.6 & -1.2 & 4 \\ 1.2 & 0.6 & -3 \end{bmatrix} \otimes \begin{bmatrix} 1 \\ 0 \end{bmatrix} \\[4pt]
&= \begin{bmatrix} 0.6 & -1.2 & 4 \\ 0 & 0 & 0 \end{bmatrix}.
\end{aligned}
\tag{15}
$$

In the next step, we calculate the final $Cx$, which is the same as $Cx_{Step1}$

$$
\begin{aligned}
Cx_{Step1} &= W_1{}^T \bullet Cx_{Step0} \times TR_{out} \\[4pt]
&= \begin{bmatrix} 0.4 & 0.6 \end{bmatrix} \bullet \begin{bmatrix} 0.6 & -1.2 & 4 \\ 0 & 0 & 0 \end{bmatrix} \times 1 \\[4pt]
&= \begin{bmatrix} 0.24 & -0.48 & 1.6 \end{bmatrix}.
\end{aligned}
\tag{16}
$$

From Eq. (16), we can determine the final output $y$ and the contributions of the input values.

$$C_{x1} = 0.24, C_{x2} = -0.48, \; C_{x3} = 1.6$$

$$y = \sum_{i=1}^{3} C_{x_i} = 1.36.$$

Equations (15) and (16) can be extended to deeper neural networks using a feedforward approach.

## Algorithm for proposed method

In this section, we describe the formal NNexplainer algorithm for calculating the contributions of the input values. For simplicity, we assume the following functions:

diag(): transform a list to diagonal matrix

mat_mul(): matrix multiplication of two matrix

**Algorithm 1  Calculation of contribution of input values.**

```
1    Inputs:
2        test_input                                    // input data for prediction
3        weights                          // list of weight matrixes of predictive model
4        N                                                    // number of layers
5    Output:
6        cont_list                      // list of contribution values for each feature
7
8    BEGIN
9
10        // build a list of transformation rate for each layer
11        TR = []                                          // transformation rate
12        FOR ln FROM 0 To (N-1)
13            calculate weighted sum WS of layer ln
14            calculate layer_outputs of layer ln          // layer_outputs = φ(WS)
15            TR_tmp := layer outputs / WS
16            append TR_tmp to TR
17        END FOR
18
19        // calculate contribution of input values
20        cont_matrix_before := diag(test_input)           // diagonal of test_input
21
22         FOR ln FROM 0 to (N-1)
23             // Calculate Cx for given layer
24            cont_matrix_this := mat_mul(WEIGHTS[ln].T, cont_matrix_before)
25             cont_matrix_before := cont_matrix_this ⊗ TR[ln]
26         END FOR
27
28        cont_list = cont_matrix_before
29        RETURN(cont_list)
30
31    END
```

list_mul(): multiply corresponding elements of two list and form a new list
list_sum(): summate all elements of a list

To aid in understanding Algorithm 1, an example of the actual calculation process is provided in the Supplemental Material.

**Table 1 Package list used to implement proposed NNexplainer.**

| Package | Version |
| --- | --- |
| keras | 3.5 |
| scikit-lern | 1.3.2 |
| pandas | 2.1.4 |
| numpy | 1.26.3 |

**Table 2 Variables in Boston housing dataset.**

| Variable | Description |
| --- | --- |
| crim | *Per capita* crime rate by town |
| zn | Proportion of residential land zoned for lots over 25,000 sq.ft. |
| indus | Proportion of non-retail business acres per town. |
| chas | Charles River dummy variable (1 if tract bounds river; 0 otherwise) |
| nox | Nitric oxides concentration (parts per 10 million) |
| rm | Average number of rooms per dwelling |
| age | Proportion of owner-occupied units built prior to 1940 |
| dis | Weighted distances to five Boston employment centres |
| rad | Index of accessibility to radial highways |
| tax | Full-value property-tax rate per $10,000 |
| ptratio | Pupil-teacher ratio by town |
| b | $1{,}000(Bk-0.63)^2$ where Bk is the proportion of blacks by town |
| lstat | % lower status of the population |
| medv | Median value of owner-occupied homes in $1,000's |

## Implementation and test

We implemented the proposed NNexplainer using Python 3.12 and its related packages, as listed in Table 1.

To test the implemented NNexplainer, we selected the Boston housing dataset (*Harrison & Rubinfeld, 1978*, https://www.kaggle.com/datasets/heptapod/uci-ml-datasets). This dataset is widely used to build example regression models, with the goal of predicting the exact house price. It contains fourteen variables (features), and "medv" is the target variable representing the house price. Table 2 presents these variables.

We built a regression model to predict "medv" based on the Boston housing dataset. Table 3 presents the correlation coefficients between "medv" and other variables. The variable "rm" exhibits a strong positive correlation with house prices, whereas "lstat" shows a strong negative correlation.

The architecture of the model is illustrated in Fig. 6, where the numbers in parentheses represent the number of nodes. The learning parameters of the model are listed in Table 4. The entire dataset is split into training and test sets, which are then standardized to a range of 0 to 1.

**Table 3 Correlation coefficient between "medv" and other features.**

| crim | zn | indus | chas | nox | rm | age |
|---|---|---|---|---|---|---|
| −0.388 | 0.36 | −0.484 | 0.175 | −0.427 | 0.695 | −0.377 |

| dis | rad | tax | ptratio | b | lstat | |
|---|---|---|---|---|---|---|
| 0.25 | −0.382 | −0.469 | −0.508 | 0.333 | −0.738 | |

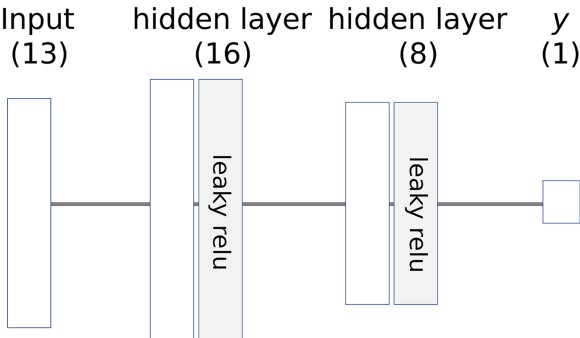

**Figure 6 Architecture of example neural network.**

**Table 4 Learning parameters to build a regression model.**

| Parameter | Value |
|---|---|
| Learning rate | 0.05 |
| Batch size | 25 |
| Epochs | 40 |
| Initializer for weight matrix | He normal |
| Activation function | Leaky relu |

## RESULTS

When the performance of the prediction model was evaluated using the test dataset, the mean absolute error (MAE) was approximately 2.33. Figure 7 presents a feature importance graph for the prediction model derived from the permutation method of the feature values (*Altmann et al., 2010*). In the model, "lstat," "dis," and "rm" are the top three features contributing to the house price, whereas the influence of "chas," "indus," and "zn" in determining the house price is minimal. The feature importance of "lstat" is 3.0, indicating that removing its influence from the prediction model would increase the MAE by 3.0.

For the prediction test, the input cases listed in Table 5 were selected. The predicted house price ("medv") for this input case is approximately 29.06, and the proposed method produces the contributions of each input value, as shown in Table 6. Figure 8 shows a bar graph corresponding to Table 5. The sum of the contributions of the input values equals the predicted house prices. From Table 5 and Fig. 8, we can observe that the values of "b"
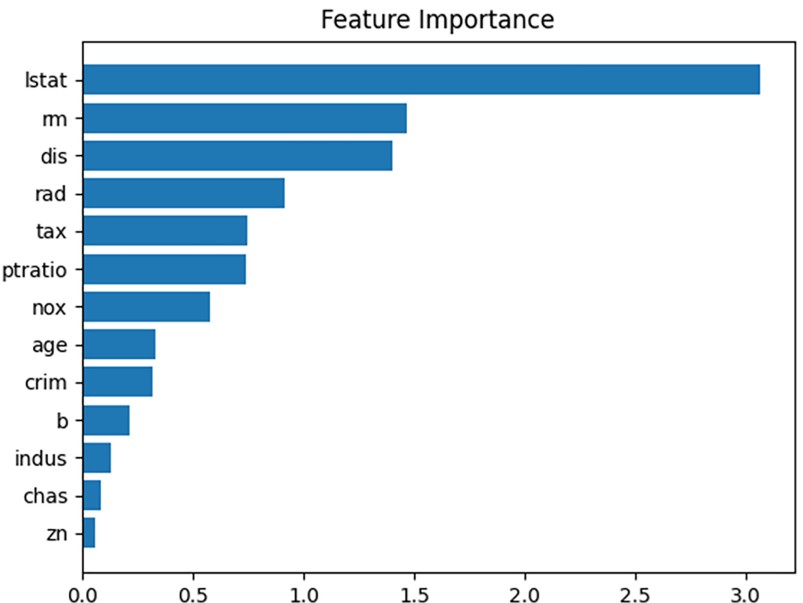

**Figure 7 Feature importance plot for given prediction model.**

**Table 5 Input case for prediction test.**

| crim | zn | indus | chas | nox | rm | age |
|------|-----|-------|------|-------|-------|------|
| 0.049 | 33 | 2.18 | 0 | 0.472 | 6.849 | 70.3 |

| dis | rad | tax | ptratio | b | lstat | |
|-------|-----|-----|---------|-------|-------|--|
| 3.183 | 7 | 222 | 18.4 | 396.9 | 7.53 | |

**Table 6 Contributions of each input value to produce 29.06.**

| crim | zn | indus | chas | nox | rm | age |
|--------|--------|-------|------|-------|-------|--------|
| −0.037 | −0.201 | 0.542 | 0 | −2.74 | 71.67 | −13.05 |

| dis | rad | tax | ptratio | b | lstat | |
|--------|-------|-------|---------|-------|-------|--|
| −10.33 | 0.357 | −4.22 | −24.98 | 20.61 | −8.56 | |

and "rm" primarily increase the house price, whereas the values of "lstat," "ptratio," "tax," "dis," "age," and "nox" decrease it.

We wish to observe the variation in the predicted value and the contributions of the input values as the specific feature values change. To achieve this, we developed an interactive contribution plot as shown in Fig. 9. The slider length represents the range of specific feature values between the minimum and maximum of that feature, allowing us to determine whether the feature value is small or large. When we move the slider for a

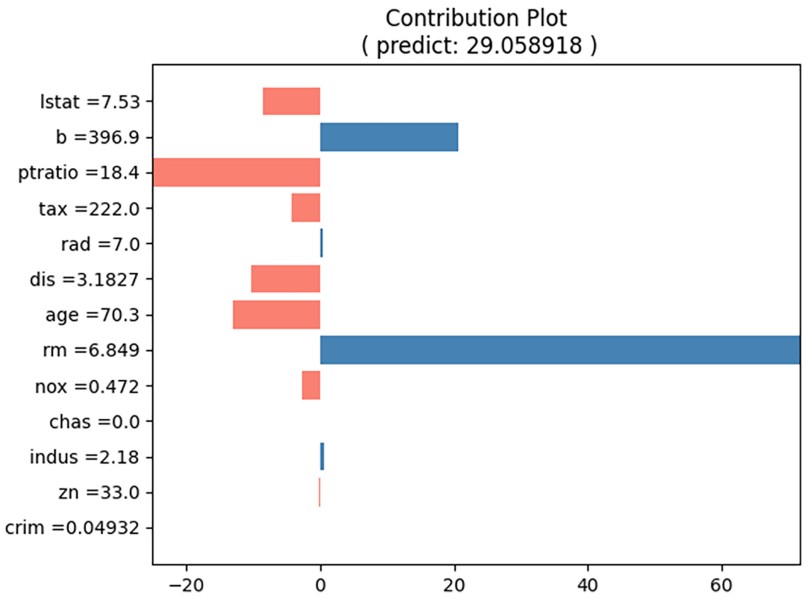

Figure 8 **Contribution plot for input values in Table 5.**

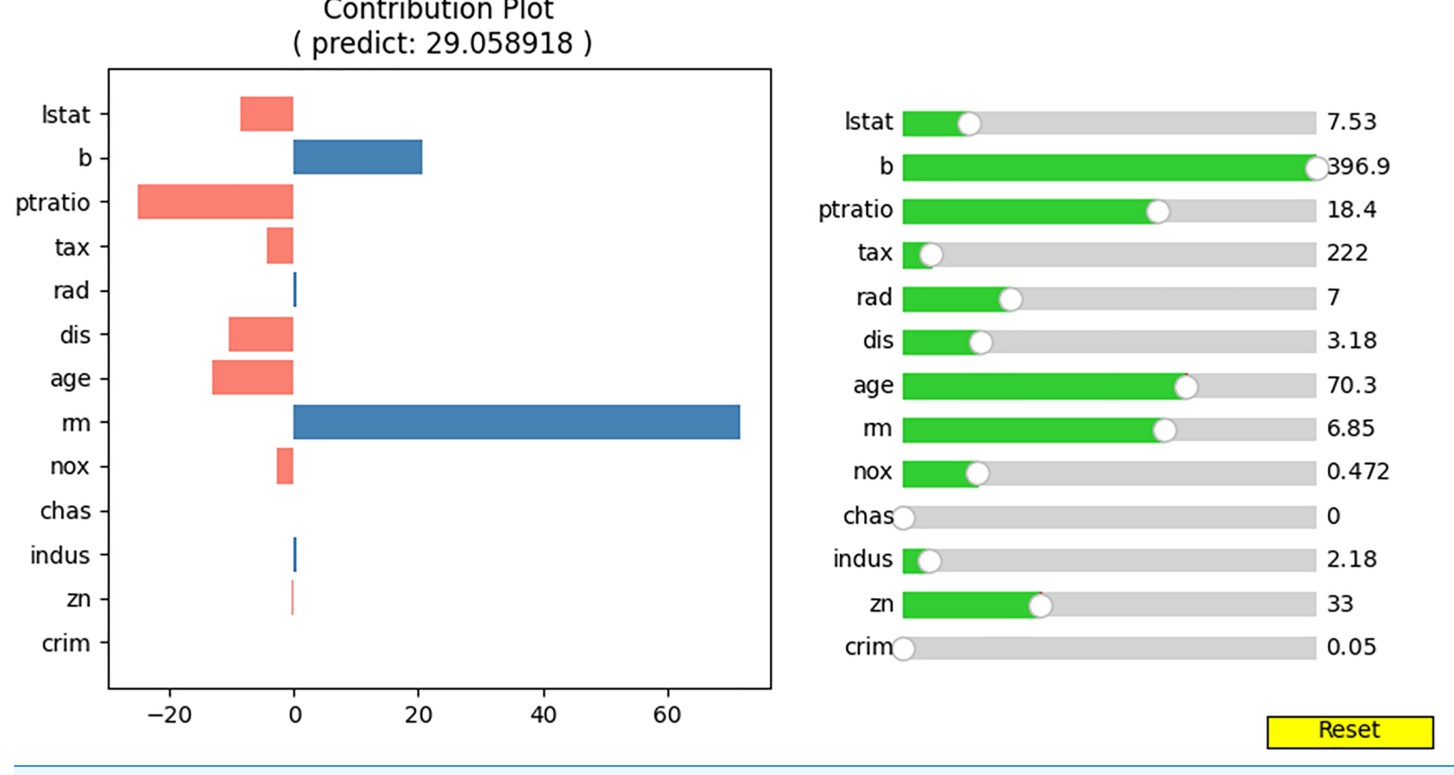

Figure 9 **Interactive contribution plot.**

feature, the feature value changes and the house price and contributions of the feature values are recalculated. Figure 10 illustrates the variance when we change the value of "lstat" from 7.53 to 18.55 *via* 13.91. From Fig. 10, we can observe several points.

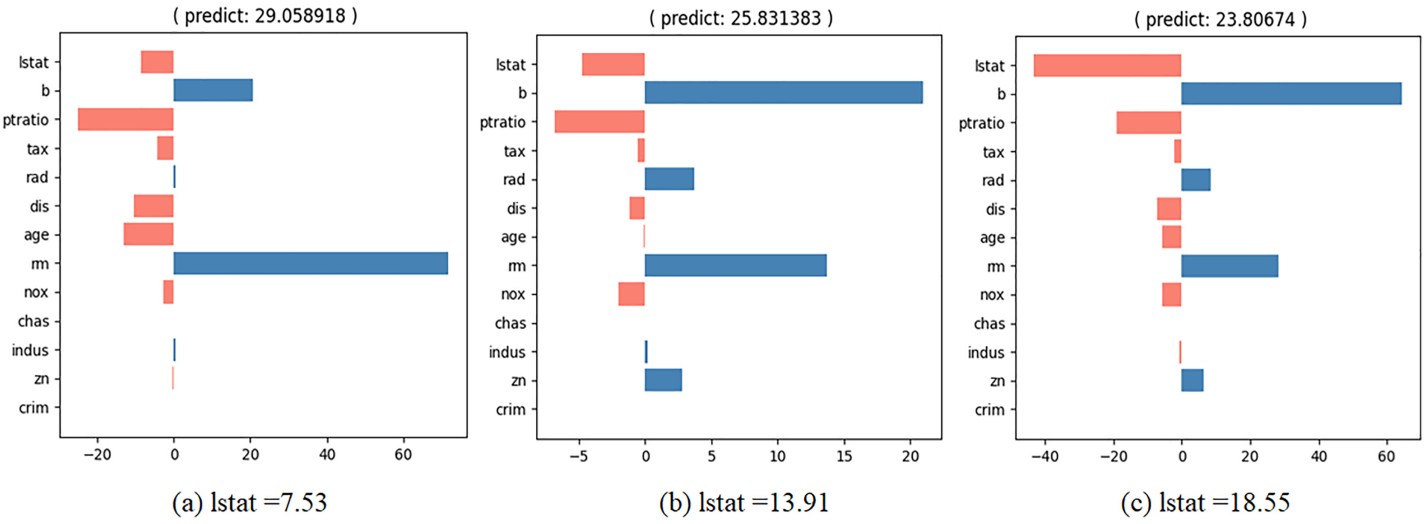

(a) lstat =7.53          (b) lstat =13.91          (c) lstat =18.55

**Figure 10 Interactive contribution plots by changing "lstat".**

- "lstat" has a negative effect on house prices. As its value increases, its relative influence on house prices also increases.
- When "lstat" increases, the relative influence of "b" also rises.
- When "lstat" increases, the relative influence of "rm" decreases.

## DISCUSSION

The proposed NNexplainer explains the prediction results generated by a neural-network-based prediction model. The prediction can be decomposed into the contributions of each input feature value. Using the interactive contribution plot, we observe the interactions among feature values. In this section, we discuss the reliability of the proposed method and then compare NNexplainer with LIME and SHAP. We also discuss the explanation for the classification model based on neural networks. Finally, we explore the application of the proposed method as a surrogate model.

### Reliability of the proposed method

As mentioned earlier, to calculate the contribution of each input variable to the prediction result, the proposed method follows the basic calculation mechanism of a feedforward neural network. In a feedforward neural network, input values move unidirectionally from the input layer to the output layer. Beyond this basic mechanism, the proposed method preserves the contribution ratio of each input value across all nodes in the hidden layer until it reaches the output node. In other words, the list of contribution ratios flows through the network and reaches the output node without exception. If the neural network is well-designed, the proposed method consistently determines the contribution of input variables to the prediction result.

In deep neural networks, the proposed method functions effectively. During the calculation process, interruptions such as the 'vanishing gradient' phenomenon do not occur. This is because, as seen in lines 18 to 35 of Algorithm 1, the contribution ratio list is updated based on the output values of each hidden layer. As long as all output values of the hidden layer are not zero, the contribution ratio list updates normally.

In conclusion, the proposed method is reliable and can be generally applied to all types of feed forward neural networks, including deep neural networks. However, additional research is needed for recurrent neural network (RNN) or convolutional neural network (CNN).

## Comparison with LIME and SHAP

To compare with LIME and SHAP, we generated LIME and SHAP plots, as shown in Figs. 11 and 12, using the Python "lime" and "shap" packages. The input feature values listed in Table 5 were tested. For clarity, we reformatted the original LIME plot using the same result data from the "lime" package. The displayed SHAP plot is a "shap waterfall" plot. In both figures, the input values are standardized versions of the original values.

In the LIME plot presented in Fig. 11, the features represented by blue bars have a positive influence on the prediction, whereas those represented by red bars have a negative influence. The values on the x-axis indicate the relative magnitude of the influence on the prediction. In the SHAP plot shown in Fig. 12, the red arrow indicates an increase in house prices, whereas the blue arrow indicates a decrease. In the SHAP model, the expected house price, expressed as $E(f(x))$ is 23.177. Each feature value can either increase or decrease the expected house price. For example, the value of "lstat" increases the expected house price by 2.14, whereas the value of "indus" decreases it by 1.02.

The contribution values displayed in the plots from LIME, SHAP, and the proposed NN explainer have different meanings; therefore, caution is necessary when interpreting these plots. The disadvantages of LIME and SHAP are discussed in the introduction. The advantages of NNexplainer over LIME and SHAP are as follows:

- NNexplainer is both simple and fast, requiring only feedforward computation for the input values. It does not require additional information beyond the weights of the trained prediction model. By contrast, LIME requires building a local interpretation model, whereas SHAP requires Monte Carlo sampling and computation of Shapley values.
- NNExplainer provides a unique and stable explanation for the given input values. The explanatory results remain consistent unless the prediction model is modified. By contrast, the explanatory results from LIME and SHAP can vary depending on the local interpretation model or the outcomes of Monte Carlo sampling.
- The contribution value for each input is absolute, similar to that of SHAP, and its meaning remains consistent across different prediction models. By contrast, LIME provides only relative contribution values.
- NNexplainer is easier to understand than LIME and SHAP. If you understand neural networks, you can also understand NNexplainer.
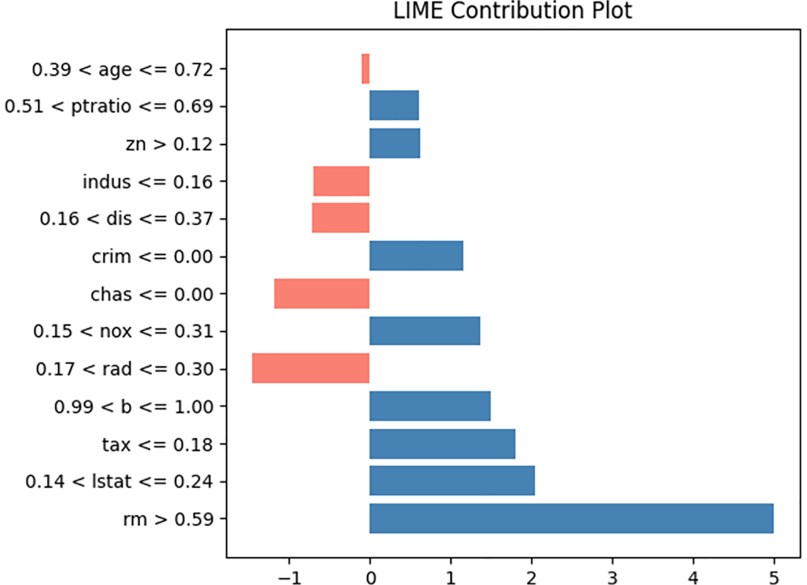

**Figure 11  LIME contribution plot for input in Table 5.**

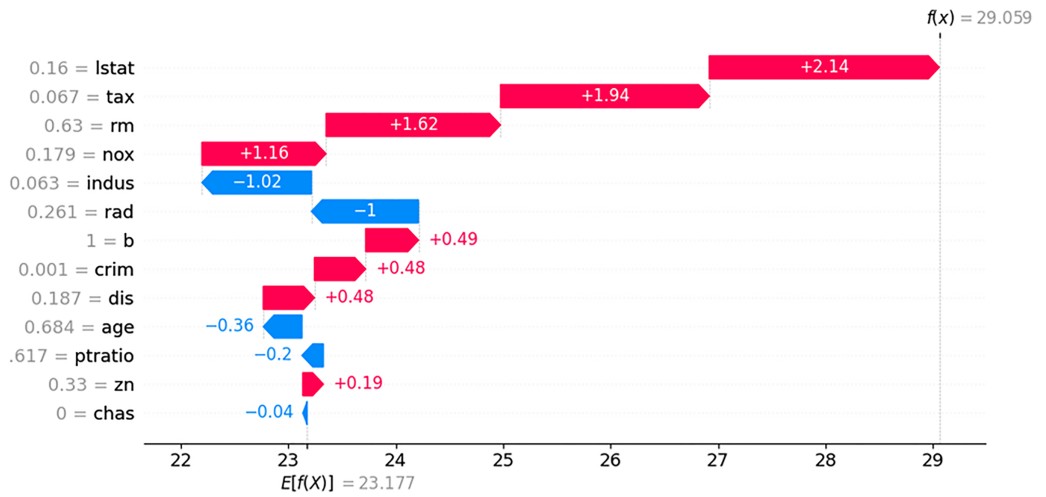

**Figure 12  SHAP contribution plot for input in Table 5.**

## Proposed method for classification model

We have discussed the proposed method from a regression perspective. However, it also applies to classification models produced by neural networks. In a neural network, the classification models have $N$ output values, where $N$ represents the number of classes, and the regression models yield a single output. Consequently, the meaning of "contribution" for each feature value differs in the proposed method. In regression models, each feature value contributes to the formation of the output value. By contrast, in classification models, each feature value supports or opposes each class. Consequently, the contribution plot for the classification model differs from that of the regression model. Figure 13 shows an

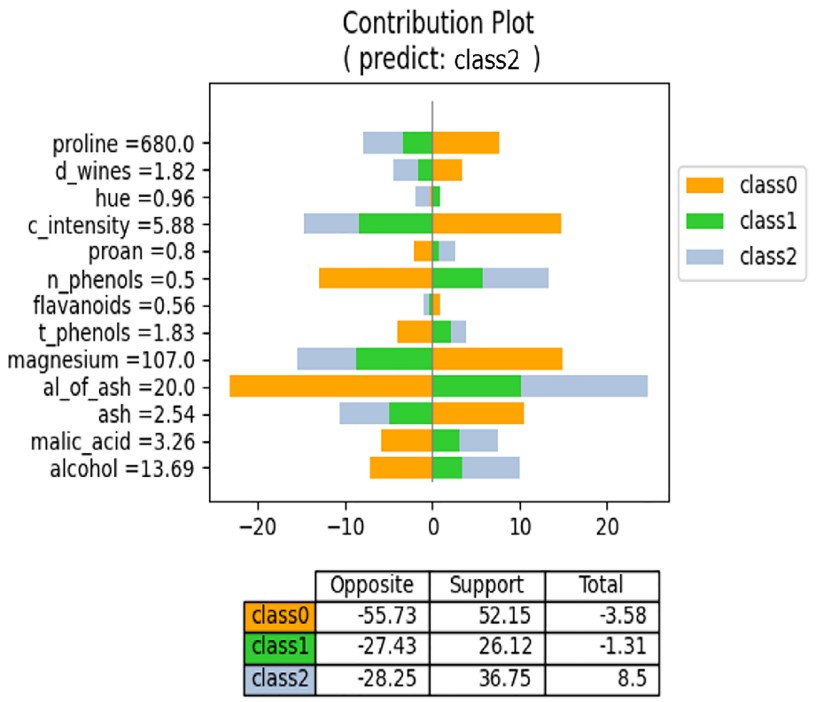

| | Opposite | Support | Total |
|---|---|---|---|
| class0 | -55.73 | 52.15 | -3.58 |
| class1 | -27.43 | 26.12 | -1.31 |
| class2 | -28.25 | 36.75 | 8.5 |

**Figure 13 NNexplainer contribution plot for classification model.**

example contribution plot for the classification model. In this task, there are three classes, and the input case is predicted to belong to Class 2. The lengths of the negative and positive side bars indicate the degree of opposition and support, respectively. For example, the value of "proline" opposes both Class 1 and Class 2 while supporting Class 0. Class 0 is strongly supported by the values of "proline," "c_intensity," and "magnesium," but is also strongly opposed by the values of "n_phenols" and "al_of_ash." Consequently, the overall degree of support is low. The prediction model selects the class with the highest support for prediction. Figure 13 illustrates the influence and direction of each feature value during the prediction process.

To obtain the contribution plot in Fig. 13, we slightly modified Algorithm 1. Lines 25–32 are not executed when layer (ln) is the final layer. In a classification model, the typical activation function for the last layer is "softmax," which transforms weighted sums into probability values ranging from 0 to 1. Consequently, the output values become very small, making it challenging to observe the influence of each feature value on the class output values.

## Proposed method as a surrogate model

The proposed NNexplainer demonstrates that regression and classification models based on neural networks are no longer black boxes. Consequently, we can explain the predictions of the black box models using neural networks and the proposed method. In other words, neural networks can serve as surrogate models to elucidate black box models. To illustrate this, we constructed a random forest (RF) regression model using the Boston

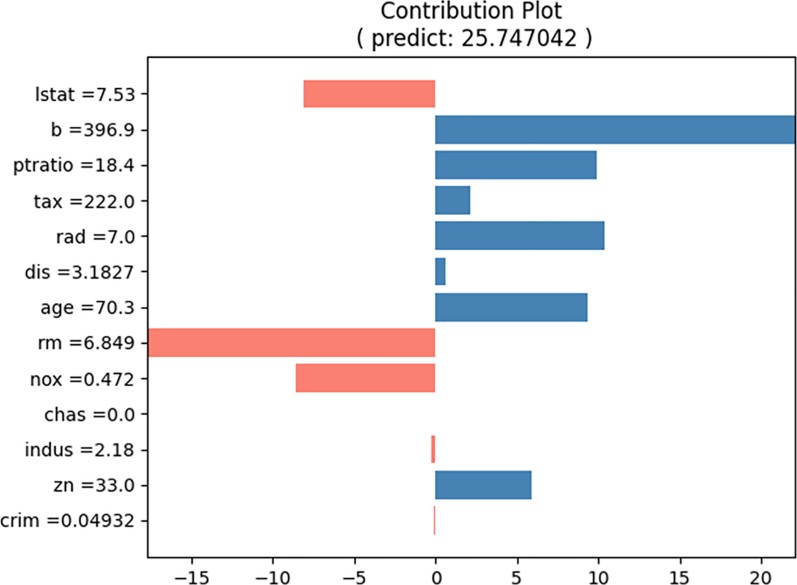

**Figure 14 Contribution plot for input in Table 5 from the surrogate model.**

housing dataset and obtained the predicted values of y_pred from the model. Subsequently, we trained a neural network model using the training set and y_pred to simulate the original RF model. The correlation coefficient between y_pred and the predictions of our surrogate model is approximately 0.97, indicating that the surrogate model effectively simulates the original RF model. Figure 14 shows the contribution plots for the input values listed in Table 5. Despite using the same input values, Figs. 8 and 14 are notably different. For instance, in the RF model, the value of "rm" decreases the house price, whereas the same value increases it in Fig. 8. This discrepancy suggests that the prediction rules or mechanisms differ between neural network and RF models. As demonstrated, the proposed method allows the neural network model to serve as a surrogate for other black box models.

### Further discussion

According to Algorithm 1, the time required to generate an explanation for a prediction result in the proposed method is proportional to the time needed to produce the prediction itself. The time required to generate a prediction in a neural network model depends on the number of hidden layers and the number of nodes in each hidden layer. Since these parameters can be arbitrarily designed, the exact time to generate a prediction result cannot be determined precisely. However, assuming the number of layers is M and each layer contains N nodes, the time complexity for generating an explanation of the prediction result is $O(N^2 \cdot (M-1))$.

The proposed model has broad applicability across various domains. For instance, in a neural network model designed to detect fraudulent transactions, it can help users understand the rationale behind classifying a transaction as fraudulent, making it easier to review or adjust decisions when necessary. Similarly, in a loan applicant credit risk

assessment model, lenders can make more informed and equitable lending decisions, even for individuals with low credit scores. In the medical field, when applied to a disease diagnosis model, the approach can provide a clear basis for identifying patients, enhancing transparency and trust in the diagnostic process.

## CONCLUSIONS

The primary significance of this study is that it provides explanations for neural-network-based prediction models, which have traditionally been considered black boxes. Currently, the proposed method is applicable only to tabular data and cannot explain input predictions in the form of images or text. In addition, global explanations, such as feature interactions, have not been explored. There is also a need to develop a wider range of interpretation tools, analogous to LIME and SHAP. These areas present opportunities for future research. The NNexplainer Python package and examples of its use are available at https://github.com/dkumango/NNexplainer/.

## ACKNOWLEDGEMENTS

I acknowledge the use of ChatGPT to refine language in the creation of this work.

### Funding

This work was supported by the Institute for Information & Communications Technology Planning & Evaluation (IITP) grand funded by the Ministry of Science, ICT (MSIT), Korea (No. RS-2023-00222191, Development of data fabric technology to support logical data integration and compound analysis of distributed data). The funders had no role in study design, data collection and analysis, decision to publish, or preparation of the manuscript.

### Grant Disclosures

The following grant information was disclosed by the authors:
Institute for Information & Communications Technology Planning & Evaluation (IITP).
Ministry of Science, ICT (MSIT), Korea: RS-2023-00222191.

### Competing Interests

The authors declare that they have no competing interests.

### Author Contributions

- Sejong Oh conceived and designed the experiments, performed the experiments, analyzed the data, performed the computation work, prepared figures and/or tables, authored or reviewed drafts of the article, and approved the final draft.

### Data Availability

The BostonHuosing dataset is available at Kaggle: https://www.kaggle.com/datasets/heptapod/uci-ml-datasets.

The Wine dataset is available at UCI and scikit:

- Aeberhard, S. & Forina, M. (1992). Wine [Dataset]. UCI Machine Learning Repository. https://doi.org/10.24432/C5PC7J.

- https://scikit-learn.org/stable/modules/generated/sklearn.datasets.load_wine.html.

The NNexplainer code is available at GitHub and Zenodo:

- https://github.com/dkumango/NNexplainer.

- dkumango. (2025). dkumango/NNexplainer: 1.0.0 (v1.0.0). Zenodo. https://doi.org/10.5281/zenodo.15025872.

## Supplemental Information

Supplemental information for this article can be found online at http://dx.doi.org/10.7717/peerj-cs.2802#supplemental-information.

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
