# Peer review of "A method for explaining individual predictions in neural networks"

_PeerJ Computer Science, doi:10.7717/peerj-cs.2802_

## Round 0.1 · original submission · Major Revisions

All 3 reviewers have significant comments which you must address.

Reviewer 1 ·

Basic reporting

Ensure all figures have consistent fonts and add more observations to captions, especially for Figure 10, to help readers understand the results better.

Consider adding more comments or a visual flowchart for Algorithm 1 to make it easier to understand.

Experimental design

The authors claim NNexplainer is more stable than LIME or SHAP (Lines 377-384). Including statistical evidence (e.g., variability across multiple runs or datasets) would strengthen this assertion.

While the Boston Housing dataset is suitable for regression, including additional datasets (e.g., classification tasks) would provide a more robust evaluation.

Validity of the findings

Discuss how your method can be used in real-world applications, like healthcare or finance, to make it more relatable.

Annotated reviews are not available for download in order to protect the identity of reviewers who chose to remain anonymous.
Cite this review as

·

Basic reporting

This paper introduces a novel method for transforming predictive neural networks from black-box models into interpretable ones and compares its performance with popular methods such as LIME (Local Interpretable Model-agnostic Explanations) and SHAP (SHapley Additive Explanations).

In lines 56 to 57, “Although machine learning has achieved considerable success in recent years, prediction models generated by machine learning algorithms remain a “black box”.” – It would be helpful for readers if the first paragraph clearly distinguishes between the two main types of prediction algorithms in machine learning: black-box algorithms (primarily neural network-based) and interpretable (or white-box) algorithms (e.g., regression models and k-NN).
In the Discussion section, it would be helpful to include more details on how NNexplainer performs regarding robustness - how well it generalizes and if it is stable.
It would be helpful to include the time complexity of the NNexplainer algorithm and if it is efficient enough to scale with bigger datasets.
In addition, the accuracy of the explanations of the individual predictions should also be studied using more in-depth methods.
The paper maintains a clear structure and provides a comprehensive list of literature studies. Thanks for providing the GitHub repository for the NNexplainer.
Some minor typos and grammatical suggestions for improvement:
In line 253, “tramsform a list to diagonal matrix” should be “transform…”.
In line 3 of Algorithm 1 table, the plural noun of “matrix” in “list of weight matrixs of predictive model” should either be matrices or matrixes. If there is only one weight matrix, it is ok to keep it as it is.
In line 24 and line 32 of the same Algorithm 1 table, “give layer” should be “given layer”.
In “Table 1: Package list used to implement proposed NNexplainer”, the package name should be “scikit-learn” instead of “scikit-lern”.
In addition, the paper title “An explanation method for individual prediction in Artificial Neural Network-based prediction model” seems a bit wordy and lacks clarity, especially the “Artificial Neural Network-Based Prediction Models” part if the paper solely focuses on prediction models in artificial neural networks.

Experimental design

The experimental design and algorithm are simple and easy to understand, especially with the Boston housing dataset. However, the paper could benefit from more detailed accuracy studies and discussions. For more details please refer to the “Basic Reporting” review section.

Validity of the findings

The findings of the paper are valid as it is supported by experiments conducted on open datasets. Please address the comments mentioned in the previous sections.

Additional comments

Thanks for the opportunity to review the manuscript. Please address the comment above.

Reviewer 3 ·

Basic reporting

All comments are included in detail in the last section.

Experimental design

All comments are included in detail in the last section.

Validity of the findings

All comments are included in detail in the last section.

Additional comments

Review Report for PeerJ Computer Science
(An explanation method for individual prediction in Artificial Neural Network-based prediction model)

1. This paper summary provides a brief summary of the Explanation Method study's background, methods used, and results. However, the abstract section needs to be detailed further.

2. Within the scope of the study, a method that can be applied in the Python library and independently of the number of nodes in the hidden layers of artificial neural networks has been proposed.

3. In the introduction section, SHAP chart, LIME chart and the importance of the subject have been mentioned. The literature in this section is very limited and needs to be detailed.

4. In the materials and methods section, a neural network, activation and neural network, and the proposed method have been mentioned as a basis. Although the dataset specified in Table-2 is deprecated and its use is at a certain level, it is suggested to use different datasets.

5. It should be explained how the parameters specified in Table-4 are determined and whether different experiments have been made. The parameters listed in Table-4 play a crucial role in the methodology and should be explained in more detail. Specifically, how these parameters were selected, the criteria used for their determination, and their impact on the model’s performance should be clarified. Furthermore, it should be mentioned whether different experiments were conducted with various parameter settings and what the results of those experiments were.

6. The architecture in Figure-6 needs to be explained more clearly and how it is determined needs to be clarified. The rationale behind the design of this architecture, the factors considered during the design process, and the selection of specific parameters should be elaborated upon.

7. The results seem appropriate at a certain level and in terms of the proposed method. However, further analysis is needed to assess the reliability and generalizability of these results.

In conclusion, the study presents an interesting approach with potential contributions to the field. However, the sections mentioned above need to be elaborated on in greater detail.

Cite this review as

---

## Round 0.2 · accepted · Accept

Dear authors, we are pleased to verify that you meet the reviewer's valuable feedback to improve your research.
Thank you for considering PeerJ Computer Science and submitting your work.

Kind regards
PCoelho

Reviewer 1 ·

Basic reporting

The revised paper looks good for publication. All of my comments have been addressed.

Experimental design

no comment

Validity of the findings

no comment

Additional comments

no comment

Cite this review as

·

Basic reporting

All the questions I raised in the previous review round have been answered. I appreciate the thorough responses and the clarity provided in the response document.

Experimental design

Same as above.

Validity of the findings

The findings of the paper are valid as they are supported by experiments conducted on open datasets.

Reviewer 3 ·

Basic reporting

All comments have been added in detail to the last section.

Experimental design

All comments have been added in detail to the last section.

Validity of the findings

All comments have been added in detail to the last section.

Additional comments

Review Report for PeerJ Computer Science
(A Method for Explaining Individual Predictions in Neural Networks)

The revisions made and the responses to the reviewer comments are sufficient. I recommend that the paper be accepted.

Cite this review as